# Design, Synthesis, Herbicidal Activity, and Structure–Activity Relationship Study of Novel 6-(5-Aryl-Substituted-1-Pyrazolyl)-2-Picolinic Acid as Potential Herbicides

**DOI:** 10.3390/molecules28031431

**Published:** 2023-02-02

**Authors:** Tong Feng, Qing Liu, Zhi-Yuan Xu, Hui-Ting Li, Wei Wei, Rong-Chuan Shi, Li Zhang, Yi-Ming Cao, Shang-Zhong Liu

**Affiliations:** 1Innovation Center of Pesticide Research, Department of Applied Chemistry, College of Science, China Agricultural University, Beijing 100193, China; 2Key Laboratory of National Forestry and Grassland Administration on Pest Chemical Control, China Agricultural University, Beijing 100193, China

**Keywords:** AFB5, docking, picolinic acid, synthesis, synthetic auxin herbicides, 3D-QSAR

## Abstract

Picolinic acid and picolinate compounds are a remarkable class of synthetic auxin herbicides. In recent years, two new picolinate compounds, halauxifen-methyl (Arylex^TM^ active) and florpyrauxifen-benzyl (Rinskor^TM^ active), have been launched as novel herbicides. Using their structural skeleton as a template, 33 4-amino-3,5-dicholor-6-(5-aryl-substituted-1-pytazolyl)-2-picolinic acid compounds were designed and synthesized for the discovery of compounds with potent herbicidal activity. The compounds were tested for inhibitory activity against the growth of *Arabidopsis thaliana* roots, and the results demonstrated that the IC_50_ value of compound **V-7** was 45 times lower than that of the halauxifen-methyl commercial herbicide. Molecular docking analyses revealed that compound **V-7** docked with the receptor auxin-signaling F-box protein 5 (AFB5) more intensively than picloram. An adaptive three-dimensional quantitative structure–activity relationship model was constructed from these IC_50_ values to guide the next step of the synthetic strategy. Herbicidal tests of the new compounds indicated that compound **V-8** exhibited better post-emergence herbicidal activity than picloram at a dosage of 300 gha^−1^, and it was also safe for corn, wheat, and sorghum at this dosage. These results demonstrated that 6-(5-aryl-substituted-1-pyrazolyl)-2-picolinic acid compounds could be used as potential lead structures in the discovery of novel synthetic auxin herbicides.

## 1. Introduction

The world’s population is continuing to grow and is expected to reach 8 billion in November 2022 [1]. Demand for food is also increasing but arable land increase is far below the need of food from population increase [2]. Ensuring the unit production of plants conducting photosynthesis in agricultural practice is one of the key measures to meet the food demand [3]. The feeding and growth of agriculture pests in cultivated land ecological systems could harm crop growth and reduce crop productivity. For instance, weeds compete with crops for light, water, and nutrients, and influence crop growth and productivity. Several measures have been taken to combat agricultural pests, such as insect pests, fungi, and weeds; among them, chemical control is the most economic and effective method. Synthetic herbicides play an important role in weed control and crop yield enhancement; however, the large-scale and long-term application of some herbicides compels weeds to generate resistance, which requires the continuous discovery of new herbicidal molecules with low resistance, low toxicity, and high efficiency [4]. Synthetic auxin herbicides with structures of phenoxyacetic acid, benzoic acid, pyridinoxyacetic acid, pyridinecarboxylic acid, and 6-aryl-2-pyridinecarboxylate are important chemical herbicides; from the international herbicide-resistant weed database [5], the number of weed species that are resistant to synthetic auxin herbicides is significantly increasing at a slower rate than others [6] because of their unique mode of action and specific binding sites in target proteins, indicating that they have great potential for the development of new herbicides. In 2007, Tan et al. presented the crystal structure of the *Arabidopsis* transport inhibitor response 1 (TIR1)-ASK1 complex and established the first structural model of a plant hormone receptor and the mode of binding between auxins and a target protein [7]. Their study guides researchers in exploring the mode of action of synthetic auxin compounds and designing new computer-aided auxin molecules. Studies have reported that 2-picolinic acid synthetic auxins herbicides have physiological functions similar to those of IAA, 2,4-D, and other auxin analogs [8,9]; however, they bind to AFB5 rather than TIR1, which is a binding protein of IAA [9,10,11,12].

Picloram and clopyralid were commercialized as herbicides in the 1960s and 1975, at application rates of 125–1120 and 105–500 gha^−1^, respectively. Subsequently, aminopyralid, discovered by modifying picloram, was commercialized as a herbicide in 2006 with application rates of 5–120 gha^−1^ [13]. In 2015, Jeffrey B. Epp et al. reported that 6-aryl-2-picolinates exhibit excellent herbicidal activities by replacing the chlorine atom with a phenyl group at position 6 of 2-picolinic acid herbicides, and they discovered two novel picolinate herbicides, halauxifen-methyl and florpyrauxifen-benzyl [13,14,15]. Even though these herbicides act on complex auxin-binding proteins, weeds inevitably generate resistance with long-term extensive application; for instance, some weeds were observed to be resistant to picloram [16,17]. In this work, we attempted to modify the chemical structure of picloram to obtain highly effective herbicidal molecules.

In 2021, Yang et al. obtained 3-chloro-6-pyrazolyl-2-picolinic acids and their ester derivatives by modifying clopyralid using substituted pyrazole rings [18]. Bioassay tests indicated that compound **c5** exhibited better postemergence herbicidal activity and broader herbicidal activity at a dosage of 400 gha^−1^ than clopyralid. This indicates that the introduction of pyrazolyl at position 6 of 2-picolinic acid could be a potential strategy for discovering a novel synthetic auxin herbicide (Figure 1).

Pyrazoles are aromatic five-membered heterocyclic ring molecules, and their structural characteristics have garnered considerable attention among researchers for the incorporation of pyrazolyl with different substituents into various structures. Therefore, they exist in a large number of biologically active molecules relevant to the pharmaceutical and agrochemical industries [19]. To date, some molecules containing pyrazole have been launched as herbicides, such as benzofenap, pyrazoxyfen, and cypyrafluone. Meanwhile, molecules containing pyrazole have exhibited potential bioactivity in recent research and patents [20,21,22,23] (Figure 2).

Inspired by the discovery of 6-aryl-2-picolinate herbicides, we designed and synthesized 33 4-amino-3,5-dichloro-6-pyrazolyl-2-picolinic acids with a phenyl-substituted pyrazole replacing the chlorine atom at position 6 of picloram, in order to explore a new herbicidal molecule (Figure 3). The inhibition of Arabidopsis thaliana root growth, herbicidal activities, and crop selectivity were tested, and the quantitative structure–activity relationship (QSAR), molecular docking, and mode of action were also preliminarily explored.

## 2. Results and Discussion

### 2.1. Chemistry

The general synthetic procedure is illustrated in Figure 1. All materials were commercially available. Intermediate **II** was prepared via a nucleophilic substitution reaction in which the chlorine atom at position 6 of picloram was replaced by hydrazine hydrate [24]. Intermediate **III** was obtained via a Claisen reaction between ethyl acetate or ethyl di/trifluoroacetate and methyl ketones [25]. Moreover, intermediate **IV** was synthesized via the Knorr cyclization reaction of intermediate **II** and intermediate **III [26]**. Owing to the unsymmetrical intermediate **III**, two region isomers are often obtained in this reaction, and 5-aryl-pyrazolyl substituted product is 10 times more abundant than 3-aryl-pyrazolyl substituted product when R_1_ is an electron-withdrawing group (R_1_ = CHF_2_; and CF_3_). A possible reason for this is that the carbon atom on the carbonyl group connecting R_1_ is a more deficient electron and is more attractive to the nitrogen atom containing the lone electron pair in 6-hydrazinyl-2-picolinitrile. When R_1_ was a non-substituted alkyl group (R_1_ = Me), the regio-selectivity weakened and the ratio of the 5-aryl-pyrazolyl substituted product to 3-aryl-pyrazolyl substituted product was in the range 3:1–5:1. Finally, the cyano group in intermediate **IV** was hydrolyzed as a carboxylic acid group to yield the target compound **V [27]**. All the target compounds were characterized through HRMS and NMR, and their NMR spectra and data are shown in the Appendix A.

### 2.2. Docking Analysis

Molecular docking was used to predict the binding modes and molecular interactions of compound **V** by MOE (Version 2020.09). The binding energy was predicted based on the structure and configurations of the compounds, as summarized in Table 1. The binding energies of almost all target compounds were less than that of picloram, which indicated that most of target compound **V** exhibited a higher affinity for AFB5. In particular, the binding energy of compound **V-7** (−8.59 kJ mol^−1^) was the lowest.

As shown in Figure 4, compound **V-7**, whose binding energy was −8.33 kJ mol^−1^, exhibited hydrogen bonding with five amino acid residues: Arg449, Arg482, Arg123, Phe127, and Asp126, whereas picloram with a binding energy of −6.53 kJ mol^−1^ exhibited hydrogen bonding with three amino acid residues: Arg449, Val485, and Leu450, which probably explained the difference of the binding energies between two molecules at a certain level. In particular, the nitrogen atom at position 2 of the pyrazolyl ring in compound **V-7** forms a hydrogen bond with residues Arg123 and Arg 482, which demonstrates that the proposed design improved the modification of molecules.

As reported by Jeffrey B. Epp et al., halauxinfen-methyl is metabolized into halauxifen in plants [13]. In comparison with the binding to AFB5, as shown in Figure 5b, compound **V-7** and halauxifen overlapped well in the same position, and compound **V-7** not only exhibited more hydrogen bonding, but also formed hydrophobic interactions between the benzene ring and the nearby phenylalanine because of the lengthening effect of the pyrazolyl ring.

### 2.3. A. thaliana Root Growth Assays to Quantify Compounds Activity

*A. thaliana* is frequently used as a model plant to explore the preliminary effects of novel compounds on plants [28], including phenotype and physiological indexes, and it can also be used to study the mechanism of plant responses to chemicals at the protein and gene levels. In this study, all target compounds were tested against *A. thaliana* root growth in at least five concentrations for preliminary biological activity and the calculation of IC_50_ values. As summarized in Table 2, some compounds exhibited better inhibition than the picloram and halauxifen-methyl commercial herbicides. In particular, compounds **V-2** and **V-7** exhibited approximately 30- and 50-fold lower IC_50_ values than the commercial herbicide halauxifen-methyl, respectively, indicating that the proposed strategy for modifying picloram is probably successful in inhibiting *A. thaliana* root growth.

Combined with the score and *p*IC_50_, as shown in Figure 6, the scatter plots of score and *p*IC_50_ are generally fitted to a line, indicating that the docking model was well predicted.

### 2.4. Three-Dimensional Quantitative Structure–Activity Relationship (3D-QSAR)

Combined with the determination of IC_50_ values against *A. thaliana* root growth inhibition and compound structure, a 3D-QSAR model was constructed using the CoMFA strategy. The model was generated with all possible combinations of steric and electrostatic fields for CoMFA [29]. To build the model, the cross-validated partial least squares (PLS) method was used, which provides a leave-one-out cross-validated correlation coefficient q^2^ and determination coefficient r^2^. Based on the basic requirements of the criteria proposed by the Golbraikh and Tropsha conditions, the model satisfying q^2^ > 0.5 and r^2^ > 0.6 is considered to be acceptable and predictive.

As listed in Table 3, the CoMFA model satisfied the q^2^ > 0.5 and r^2^ > 0.6 criteria. This model exhibited a q^2^ of 0.679; r^2^ of 0.848; standard error of estimate (SEE) of 0.337; Fisher test value (F) of 44.660; and an optimum number of components (ONC) of 4. The contributions of steric and electrostatic fields were 62.7 and 36.3%, respectively.

#### 2.4.1. Scatter Plots

As shown in Figure 7, the scatter plots of the actual and predicted inhibitory activities of the training set compounds fit well to a line, indicating that the constructed CoMFA model exhibits predictive capability.

#### 2.4.2. Contour Map Analysis

The CoMFA contour map was created using the StDev*Coeff mapping option, which allows visualization of each field effect. Compound **V-7**, which had the lowest IC_50_ value, was superimposed on the contour map. The contributions to the favorable and unfavorable regions in each field were 80 and 20%, respectively.

In the steric contour map shown in Figure 8a, the green contour block indicates that the presence of bulky steric groups in this area would be favorable for the biological activity of a compound, whereas the yellow block indicates that the bulky groups would be unfavorable. The green contour blocks appear in four parts around compound **V-7**, while the largest green contour block appears at the fourth position of the benzene ring connected to the pyrazole ring, which indicates the reason for the different activities between compounds **V-7** and **V-1**. Their substituents are chlorine atoms and methyl groups, respectively, and chlorine atom groups exhibit larger occupation than methyl groups. In addition, it can be observed in this figure that a small yellow contour block appears next to the largest green contour block, indicating that the bulky groups in this area are unfavorable. This could explain why compound **V-7** was more active than compound **V-32**, because in comparison with chlorine atoms, the *tert*-butyl group occupies a larger space, resulting in its invasion of the area of the yellow contour block.

As shown in Figure 8b, the blue contours indicate that the electropositive groups are favorable for increasing the activity of a compound, whereas the red contours indicate that the electropositive groups are unfavorable for the activity of a compound. On the side of the carbonyl group, a large red contour block appears, indicating the importance of the carbonyl group at position 2 of the pyridine. The blue contour block next to the trifluoromethyl group at position 3 of the pyrazolyl ring explains why compound **V-7** exhibits higher activity owing to the presence of electronegative substituents in the blue contour region.

### 2.5. Greenhouse Activity Assay

Based on the design strategy and docking analysis, the herbicidal activities of the new compounds were tested in a greenhouse against six common weeds, including three gramineous weeds: *Setaria glauca* (SG), *Digitaria sanguinalis* (DS), and *Echinochloa crusgalli* (EC); three broadleaf weeds: *Chenopodium album* (CA), *Abutilon theophrasti* (AT), and *Amaranthus retroflexus* (AR); and two commercial herbicides, picloram and halauxifen-methyl, were selected as the control groups. As summarized in Table 4 and shown in Figure 9, most of the compounds exhibited post-emergence inhibitory activities against the weeds mentioned above. Generally, the inhibitory activities of almost all compounds on broadleaf weeds are stronger than those on gramineous weeds, which is similar to the inhibitory activities of picloram and halauxifen-methyl. This also indicates that the target compounds may have similar mechanisms of action as commercial herbicides. Furthermore, compounds **V-1**–**V-18** with halogen-substituted phenyl exhibited better herbicidal activities than compounds **V-19**–**V-33** with alkyl-substituted phenyl. Notably, compound **V-8** at a dosage of 300 gha^−1^ exhibits 100, 100, and 95% post-emergence injury values against CA, AT, and AR, respectively, and 100, 40, and 100% pre-emergence injury values against CA, AT, and AR, respectively. This illustrates that the herbicidal activity of compound **V-8** was better than that of picloram. Furthermore, there was a slight variance between the herbicidal activities of compounds **V** and inhibition of *A. thaliana* root growth, possibly because of the AFB5 difference in *A. thaliana* and the tested weeds.

Based on these results, the herbicidal activity of compound **V-8** was tested on broadleaf weeds at dosages of 300, 150, and 75 gha^−1^. As summarized in Table 5, the inhibition of compound **V-8** against the weeds gradually decreased with decreasing dosages in the pre-emergence test; however, the decreasing trend was small. In the post-emergence test, the inhibition of compound **V-8** remained high at a dosage of 150 gha^−1^ compared with that at a dosage of 300 gha^−1^. These results indicated that compound **V-8** exhibited a better herbicidal effect than picloram.

To further investigate the crop selectivity of compound **V-8**, three common crops—including corn, wheat, and sorghum under pre-emergence and post-emergence conditions—were treated. As summarized in Table 6, compound **V-8** was safe for wheat while slightly damaging corn and sorghum under pre-emergence conditions; however, it was completely safe for the three crops under post-emergence conditions.

In summary, these results demonstrated that the target compounds are potential candidates for herbicide discovery research, and they need to be studied further.

## 3. Materials and Methods

### 3.1. Chemicals and Instruments

All the reagents and solvents were purchased from commercial suppliers (Beijing InnoChem Science & Technology Co., Ltd., Beijing, China and Sinopharm Chemical Reagent Co., Ltd., Beijing, China). All reactions were monitored using thin-layer chromatography (TLC) run on silica gel glass plates (Qingdao Broadchem Industrial, Qingdao, China). ^1^H and ^13^C NMR spectroscopy were recorded using a Bruker AM-500 spectrometer with temperature control at 21–23 °C, using DMSO-*d6* or CDCl_3_ as the solvent and tetramethyl silane (TMS) as the internal reference. In the spectra, the chemical shifts (δ) were given in parts per million (ppm). High-resolution mass spectra (HRMS) were determined with an Agilent 6540 QTOF instrument.

### 3.2. Synthesis

#### 3.2.1. General Synthetic Procedure of Intermediate **II**

In a 500 mL, three-necked, round-bottom flask, compound **I** (100 mmol) and anhydrous potassium carbonate (200 mmol) were added to tetrahydrofuran (250 mL). Subsequently, 80% hydrazine hydrate (300 mmol) was slowly added to the reaction mixture while stirring at 0 °C. After the addition, the reaction mixture was heated to 75 °C in an oil bath for 6 h of reflux. Once the reaction was complete, the mixture was cooled to room temperature and filtered. The obtained solid was washed with water to obtain intermediate **II** as an off-white solid (yield 73%): ^1^H NMR (300 MHz, DMSO-*d6*) δ 9.63 (s, 1H), 7.18 (s, 2H), 4.53 (s, 2H).

#### 3.2.2. General Synthetic Procedure of Intermediate **III**

Two different procedures were used to synthesize 1,3-diketones in this study. (1) When R_1_ was a methyl group, ethyl acetate was used as the reactant and solvent. In a 100 mL round-bottom flask, 60% sodium hydride (28.8 mmol) was slowly added to 30 mL of ethyl acetate under stirring at −5 °C. Thereafter, a mixture of methyl ketones (28.8 mmol) and 10 mL of ethyl acetate was added slowly to the reaction solution. After addition, the entire system was warmed to room temperature and stirred for 6 h. The reaction was quenched with aqueous hydrochloric acid (1 N, 30 mL), acidified to a pH range of 1–2, and subsequently extracted using ethyl acetate (3 × 15 mL). The combined organic phases were dried over anhydrous sodium sulfate and concentrated under a vacuum. The residue was purified via flash column chromatography (n-hexane) to obtain intermediate **III** (yields 65.8–78.4%). (2) When R_1_ was a difluoromethyl or trifluoromethyl, the procedure was as follows: in a 100 mL round-bottom flask, 60% sodium hydride (43.2 mmol) was slowly added to 30 mL of ethyl ether with stirring at −5 °C. Thereafter, a mixture of methyl ketones (28.8 mmol) and ester (34.56 mmol) was slowly added to the reaction solution. After addition, the entire system was warmed to room temperature and stirred overnight. The reaction was quenched with aqueous hydrochloric acid (1 N, 30 mL), acidified to a pH range of 1–2, and extracted using ethyl acetate (3 × 15 mL). The combined organic phases were dried over anhydrous sodium sulfate and concentrated under vacuum. The residue was purified via flash column chromatography (n-hexane/ethyl acetate = 10:1) to afford intermediate **III** (yields 87.6–92.2%).

#### 3.2.3. General Synthetic Procedure of Intermediate **IV**

In a 50 mL round-bottom flask, intermediate **III** (5 mmol) was added to a solution of intermediate **II** (5 mmol) in 20 mL of ethanol at room temperature. Thereafter, concentrated sulfuric acid was slowly added to the stirred solution, and the reaction mixture was heated to 75 °C in an oil bath and kept at reflux for 2 h. The reaction was cooled to room temperature, quenched with a saturated sodium carbonate solution, and extracted using ethyl acetate (3 × 15 mL). The combined organic phases were dried over anhydrous sodium sulfate and concentrated under vacuum. The residue was purified via flash column chromatography (n-hexane/ethyl acetate = 6:1) to afford intermediate **IV** (yields 80.3–91.7%).

#### 3.2.4. General Synthetic Procedure of Compound **V**

Intermediate **IV** (1.067 mmol) was dissolved in 80% aqueous sulfuric acid (10 mL) in a 25 mL round-bottom flask. Thereafter, the reaction solution was heated to 100 °C in an oil bath and kept at reflux for 2 h. The reaction mixture was cooled to room temperature and quenched with water. The white solid was collected through filtration and dried to achieve the target compound **V** (yields 90.1–99.0%).

Compound **V-1** 4-Amino-3,5-dichloro-6-(3-methyl-5-(p-tolyl)-1H-pyrazol-1-yl)picolinic acid. White solid. ^1^H NMR (500 MHz, DMSO-*d6*) δ 13.90 (s, 1H), 7.21 (s, 2H), 7.14 (d, *J* = 8.1 Hz, 2H), 7.08 (d, *J* = 8.1 Hz, 2H), 6.47 (s, 1H), 2.25 (s, 3H), 2.25 (s, 3H). ^13^C NMR (126 MHz, DMSO-*d6*) δ 165.94, 150.50, 149.53, 147.69, 147.12, 145.13, 138.41, 129.79, 127.35, 127.13, 113.10, 112.26, 106.17, 21.19, 13.79. HRMS calcd. for C_17_H_14_Cl_2_N_4_O_2_ ([M-H]^−^), 375.0416; found, 375.0413.

Compound **V-2** 4-Amino-3,5-dichloro-6-(3-(difluoromethyl)-5-(p-tolyl)-1H-pyrazol-1-yl)picolinic acid. White solid. ^1^H NMR (300 MHz, DM*S*O-*d6*) δ 14.03 (s, 1H), 7.35 (s, 2H), 7.19 (d, *J* = 8.6 Hz, 2H), 7.16 (d, *J* = 8.6 Hz, 2H), 7.10 (t, *J* = 54.35 Hz, 1H), 6.99 (s, 1H), 2.27 (s, 3H). ^13^C NMR (126 MHz, DMSO-*d6*) δ 165.70, 163.80, 161.84, 150.85, 148.00, 147.78, 147.54, 147.30, 146.49, 145.07, 130.24, 130.17, 125.32, 116.60, 116.43, 113.53, 113.01, 112.72, 111.67, 109.82, 104.16. HRMS calcd. for C_17_H_12_Cl_2_F_2_N_4_O_2_ ([M-H]^−^), 411.0227; found, 411.0239.

Compound **V-3** 4-Amino-3,5-dichloro-6-(5-(p-tolyl)-3-(trifluoromethyl)-1H-pyrazol-1-yl)picolinic acid. White solid. ^1^H NMR (500 MHz, DM*S*O-*d6*) δ 14.04 (s, 1H), 7.40 (s, 2H), 7.25 (s, 1H), 7.21 (d, *J* = 8.9 Hz, 2H), 7.18 (d, *J* = 8.9 Hz, 2H), 2.28 (s, 3H). ^13^C NMR (126 MHz, DMSO-*d6*) δ 165.73, 150.83, 147.51, 146.63, 146.38, 143.22, 142.92, 142.62, 142.32, 139.77, 130.04, 127.85, 125.32, 124.88, 122.74, 120.60, 118.46, 113.17, 112.73, 104.51, 40.30, 21.22. HRMS calcd. for C_17_H_11_Cl_2_F_3_N_4_O_2_ ([M-H]^−^), 429.0133; found, 429.0143.

Compound **V-4** 4-Amino-3,5-dichloro-6-(5-(4-fluorophenyl)-3-methyl-1H-pyrazol-1-yl)picolinic acid. Yellow solid. ^1^H NMR (500 MHz, DMSO-*d6*) δ 13.83 (s, 1H), 7.25 (s, 2H), 7.24–7.22 (m, 2H), 7.22–7.18 (m, 2H), 6.52 (s, 1H), 2.26 (s, 3H). ^13^C NMR (126 MHz, DMSO-*d6*) δ 165.88, 163.36, 161.41, 150.61, 149.65, 147.36, 147.15, 144.06, 129.71, 129.64, 126.50, 116.37, 116.19, 112.90, 112.32, 106.68, 13.76. HRMS calcd. for C_16_H_11_Cl_2_FN_4_O_2_ ([M-H]^−^), 379.0165; found, 379.0159.

Compound **V-5** 4-Amino-3,5-dichloro-6-(3-(difluoromethyl)-5-(4-fluorophenyl)-1H-pyrazol-1-yl)picolinic acid. Yellow solid. ^1^H NMR (500 MHz, DMSO-*d6*) δ 13.98 (s, 1H), 7.40 (s, 2H), 7.35–7.33 (m, 2H), 7.28–7.24 (m, 2H), 7.13 (t, *J* = 54.3 Hz, 1H), 7.07 (s, 1H). ^13^C NMR (126 MHz, DMSO-*d6*) δ 165.70, 163.80, 161.84, 150.85, 148.00, 147.78, 147.54, 147.30, 146.49, 145.07, 130.24, 130.17, 125.32, 116.60, 116.43, 113.53, 113.01, 112.72, 111.67, 109.82, 104.16. HRMS calcd. for C_16_H_9_Cl_2_F_3_N_4_O_2_ ([M-H]^−^), 414.9976; found, 414.9986.

Compound **V-6** 4-Amino-3,5-dichloro-6-(5-(4-fluorophenyl)-3-(trifluoromethyl)-1H-pyrazol-1-yl)picolinic acid. Yellow solid. ^1^H NMR (500 MHz, DMSO-*d6*) δ 14.07 (s, 1H), 7.46 (s, 2H), 7.38–7.36 (m, 2H), 7.35 (s, 1H), 7.31–7.27 (m, 2H). ^13^C NMR (126 MHz, DMSO-*d6*) δ 165.61, 164.01, 162.03, 150.97, 147.30, 146.07, 145.56, 143.29, 142.99, 142.69, 142.39, 130.44, 130.37, 124.81, 124.71, 124.68, 124.68, 122.67, 120.53, 118.39, 116.71, 116.54, 113.34, 112.67, 105.06. HRMS calcd. for C_16_H_8_Cl_2_F_4_N_4_O_2_ ([M-H]^−^), 432.9882; found, 432.9888.

Compound **V-7** 4-Amino-3,5-dichloro-6-(5-(4-chlorophenyl)-3-methyl-1H-pyrazol-1-yl)picolinic acid. White solid. ^1^H NMR (500 MHz, DMSO-*d6*) δ 13.50 (s, 1H), 7.43 (d, *J* = 8.55 Hz, 2H), 7.28 (s, 2H), 7.21 (d, *J* = 8.55 Hz, 2H), 6.57 (s, 1H), 2.27 (s, 3H). ^13^C NMR (126 MHz, DMSO-*d6*) δ 165.85, 150.66, 149.78, 147.28, 147.11, 143.85, 133.63, 129.34, 129.16, 128.79, 112.80, 112.39, 106.96, 13.76. HRMS calcd. for C_16_H_10_Cl_3_N_4_O_2_ ([M-H]^−^), 394.9869; found, 395.0023.

Compound **V-8** 4-Amino-3,5-dichloro-6-(5-(4-chlorophenyl)-3-(difluoromethyl)-1H-pyrazol-1-yl)picolinic acid. White solid. ^1^H NMR (300 MHz, DMSO-*d6*) δ 13.95 (s, 1H), 7.48 (d, *J* = 8.4 Hz, 2H), 7.40 (s, 2H),7.30 (d, *J* = 8.4 Hz, 2H), 7.14 (t, *J* = 54.3 Hz, 1H), 7.11 (s, 1H). ^13^C NMR (126 MHz, DMSO-*d6*) δ 165.65, 150.88, 148.08, 147.85, 147.62, 147.31, 146.41, 144.87, 134.49, 129.61, 129.54, 127.61, 113.47, 113.07, 111.62, 109.77, 104.43. HRMS calcd. for C_16_H_8_Cl_3_F_2_N_4_O_2_ ([M-H]^−^), 430.9681; found, 430.9688.

Compound **V-9** 4-Amino-3,5-dichloro-6-(5-(4-chlorophenyl)-3-(trifluoromethyl)-1H-pyrazol-1-yl)picolinic acid. White solid. ^1^H NMR (500 MHz, DMSO-*d6*) δ 14.06 (s, 1H), 7.51 (d, *J* = 8.5 Hz, 2H), 7.47 (s, 2H), 7.38 (s, 1H), 7.33 (d, *J* = 8.5 Hz, 2H). ^13^C NMR (126 MHz, DMSO-*d6*) δ 165.61, 151.00, 147.34, 146.00, 145.36, 143.36, 143.06, 142.76, 142.46, 134.90, 129.75, 129.62, 126.99, 124.77, 122.63, 120.49, 118.35, 113.39, 112.58, 105.32. HRMS calcd. for C_16_H_7_Cl_3_F_3_N_4_O_2_ ([M-H]^−^), 448.9587; found, 448.9592.

Compound **V-10** 4-Amino-3,5-dichloro-6-(5-(3-chlorophenyl)-3-methyl-1H-pyrazol-1-yl)picolinic acid. Yellow solid. ^1^H NMR (500 MHz, Chloroform-*d*) δ 7.29 (s, 1H), 7.28 (d, *J* = 7.75 Hz, 1H), 7.23 (dd, *J* = 8.0 Hz, 7.75 Hz, 1H), 6.99 (d, *J* = 7.75 Hz, 1H), 6.40 (s, 1H), 5.62 (s, 2H), 2.41 (s, 3H). ^13^C NMR (126 MHz, DMSO-*d6*) δ 165.85, 150.64, 149.79, 147.23, 143.48, 133.82, 131.85, 131.13, 128.77, 127.35, 125.91, 112.85, 112.34, 107.27, 13.77. HRMS calcd. for C_16_H_10_Cl_3_N_4_O_2_ ([M-H]^−^), 394.9869; found, 394.9873.

Compound **V-11** 4-Amino-3,5-dichloro-6-(5-(3-chlorophenyl)-3-(difluoromethyl)-1H-pyrazol-1-yl)picolinic acid. Yellow solid. ^1^H NMR (500 MHz, DMSO-*d6*) δ 13.96 (s, 1H), 7.46–7.45 (m, 1H), 7.45 (s, 1H), 7.43–7.40 (m, 1H), 7.39 (s, 2H), 7.18 (s, 1H), 7.16–7.13 (m, 1H), 7.14 (t, *J* = 54.2 Hz, 1H). ^13^C NMR (126 MHz, DMSO-*d6*) δ 165.64, 150.87, 148.10, 147.87, 147.64, 147.38, 146.39, 144.50, 134.01, 131.28, 130.68, 129.58, 127.88, 126.30, 113.45, 113.04, 112.71, 111.60, 109.75, 104.81. HRMS calcd. for C_16_H_8_Cl_3_F_2_N_4_O_2_ ([M-H]^−^), 430.9681; found, 430.9691.

Compound **V-12** 4-Amino-3,5-dichloro-6-(5-(3-chlorophenyl)-3-(trifluoromethyl)-1H-pyrazol-1-yl)picolinic acid. Yellow solid. ^1^H NMR (500 MHz, DMSO-*d6*) δ 14.01 (s, 1H), 7.52–7.39 (m, 6H), 7.15 (d, *J* = 7.7 Hz, 1H). ^13^C NMR (126 MHz, DMSO-*d6*) δ 165.58, 151.00, 147.29, 145.97, 144.98, 143.37, 143.07, 142.77, 142.47, 134.10, 131.36, 130.01, 129.95, 128.09, 126.36, 125.69, 124.74, 124.60, 122.60, 120.46, 118.33, 113.36, 112.67, 105.70. HRMS calcd. for C_16_H_7_Cl_3_F_3_N_4_O_2_ ([M-H]^−^), 448.9587; found, 448.9592.

Compound **V-13** 4-Amino-6-(5-(4-bromophenyl)-3-methyl-1H-pyrazol-1-yl)-3,5-dichloropicolinic acid. Yellow solid. ^1^H NMR (500 MHz, DMSO-*d6*) δ 13.85 (s, 1H), 7.56 (d, *J* = 8.2 Hz, 2H), 7.23 (s, 2H), 7.14 (d, *J* = 8.3 Hz, 2H), 6.57 (s, 1H), 2.27 (s, 3H). ^13^C NMR (126 MHz, DMSO-*d6*) δ 165.90, 150.62, 149.77, 147.25, 143.89, 132.26, 129.41, 129.15, 122.28, 112.67, 112.30, 106.93, 13.76. HRMS calcd. for C_16_H_10_BrCl_2_N_4_O_2_ ([M-H]^−^), 440.9344; found, 440.9348.

Compound **V-14** 4-Amino-6-(5-(4-bromophenyl)-3-(difluoromethyl)-1H-pyrazol-1-yl)-3,5-dichloropicolinic acid. Yellow solid. ^1^H NMR (500 MHz, DMSO-*d6*) δ 13.92 (s, 1H), 7.62 (d, *J* = 8.5 Hz, 1H), 7.38 (s, 2H), 7.23 (d, *J* = 8.5 Hz, 1H), 7.13 (t, *J* = 54.2 Hz, 1H), 7.11 (s, 1H). ^13^C NMR (126 MHz, DMSO-*d6*) δ 165.61, 150.89, 148.10, 147.88, 147.65, 147.26, 146.41, 144.93, 132.45, 129.83, 127.97, 123.21, 113.47, 113.10, 112.64, 111.61, 109.76, 104.43. HRMS calcd. for C_16_H_8_BrCl_2_F_2_N_4_O_2_ ([M-H]^−^), 476.9155; found, 476.9159.

Compound **V-15** 4-Amino-6-(5-(4-bromophenyl)-3-(trifluoromethyl)-1H-pyrazol-1-yl)-3,5-dichloropicolinic acid. Yellow solid. ^1^H NMR (500 MHz, DMSO-*d6*) δ 13.92 (s, 1H), 7.57 (d, *J* = 8.5 Hz, 2H), 7.39 (s, 2H), 7.31 (s, 1H), 7.18 (d, *J* = 8.5 Hz, 2H). ^13^C NMR (126 MHz, DMSO-*d6*) δ 165.58, 151.01, 147.27, 145.99, 145.42, 143.38, 143.08, 142.77, 142.48, 132.54, 129.95, 127.33, 124.76, 123.64, 122.62, 120.48, 118.34, 113.42, 112.58, 105.32. HRMS calcd. for C_16_H_7_BrCl_2_F_3_N_4_O_2_ ([M-H]^−^), 494.9061; found, 494.9066.

Compound **V-16** 4-Amino-6-(5-(2-bromophenyl)-3-methyl-1H-pyrazol-1-yl)-3,5-dichloropicolinic acid. Yellow solid. ^1^H NMR (500 MHz, DMSO-*d6*) δ 13.71 (s, 1H), 7.66 (dd, *J* = 7.9, 1.1 Hz, 1H), 7.31 (td, *J* = 7.5, 1.2 Hz, 1H), 7.26 (td, *J* = 7.7, 1.8 Hz, 1H), 7.16 (s, 2H), 7.15 (dd, *J* = 7.7, 1.8 Hz, 1H), 6.48 (s, 1H), 2.30 (s, 3H). ^13^C NMR (126 MHz, DMSO-*d6*) δ 165.79, 150.43, 149.03, 146.67, 146.64, 143.25, 133.51, 132.02, 131.26, 130.98, 127.84, 123.02, 112.09, 111.70, 109.10, 13.83. HRMS calcd. for C_16_H_10_BrCl_2_N_4_O_2_ ([M-H]^−^), 440.9344; found, 440.9351.

Compound **V-17** 4-Amino-6-(5-(2-bromophenyl)-3-(difluoromethyl)-1H-pyrazol-1-yl)-3,5-dichloropicolinic acid. Yellow solid. ^1^H NMR (500 MHz, DMSO-*d6*) δ 13.82 (s, 1H), 7.71 (dd, *J* = 7.7, 1.1 Hz, 1H), 7.38–7.27 (m, 4H), 7.22 (dd, *J* = 7.3, 2.0 Hz, 1H), 7.17 (t, *J* = 54.2 Hz, 1H), 6.98 (s, 1H). ^13^C NMR (126 MHz, DMSO-*d6*) δ 165.61, 150.66, 147.38, 147.15, 146.92, 146.85, 145.86, 144.28, 133.60, 132.27, 131.73, 129.98, 128.01, 123.25, 113.51, 112.45, 112.13, 111.66, 109.81, 106.57, 40.30. HRMS calcd. for C_16_H_8_BrCl_2_F_2_N_4_O_2_ ([M-H]^−^), 476.9155; found, 476.9158.

Compound **V-18** 4-Amino-6-(5-(2-bromophenyl)-3-(trifluoromethyl)-1H-pyrazol-1-yl)-3,5-dichloropicolinic acid. Yellow solid. ^1^H NMR (500 MHz, DMSO-*d6*) δ 13.91 (s, 1H), 7.73 (dd, *J* = 7.6, 1.5 Hz, 1H), 7.37 (m, 4H), 7.26(dd, *J* = 7.3, 2.0 Hz, 1H), 7.25 (s, 1H). ^13^C NMR (126 MHz, DMSO-*d6*) δ 165.53, 150.77, 146.84, 145.45, 144.81, 142.67, 142.37, 142.07, 141.77, 133.62, 132.37, 132.08, 129.32, 128.07, 124.80, 123.36, 122.66, 120.52, 118.38, 112.78, 112.16, 107.41. HRMS calcd. for C_16_H_7_BrCl_2_F_3_N_4_O_2_ ([M-H]^−^), 494.9061; found, 494.9064.

Compound **V-19** 4-Amino-3,5-dichloro-6-(5-(4-isopropylphenyl)-3-methyl-1H-pyrazol-1-yl)picolinic acid. Yellow solid. ^1^H NMR (500 MHz, DMSO-*d6*) δ 13.78 (s, 1H), 7.23 (s, 2H), 7.21 (d, *J* = 8.4 Hz, 2H), 7.13 (d, *J* = 8.3 Hz, 2H), 6.48 (s, 1H), 2.84 (hept, *J* = 6.9 Hz, 1H), 2.25 (s, 3H), 1.16 (d, *J* = 6.9 Hz, 6H). ^13^C NMR (126 MHz, DMSO-*d6*) δ 165.92, 150.52, 149.50, 149.12, 147.78, 147.12, 145.03, 127.36, 127.19, 113.23, 112.31, 106.24, 33.52, 24.05, 13.78. HRMS calcd. for C_19_H_17_Cl_2_N_4_O_2_ ([M-H]^−^), 403.0729; found, 403.0734.

Compound **V-20** 4-Amino-3,5-dichloro-6-(3-(difluoromethyl)-5-(4-isopropylphenyl)-1H-pyrazol-1-yl)picolinic acid. White solid. ^1^H NMR (500 MHz, DMSO-*d6*) δ 13.95 (s, 1H), 7.36 (s, 2H), 7.27 (d, *J* = 8.3 Hz, 2H), 7.22 (d, *J* = 8.3 Hz, 2H), 7.11 (t, *J* = 54.3 Hz, 1H), 7.01 (s, 1H), 2.86 (hept, *J* = 6.9 Hz, 1H), 1.17 (d, *J* = 6.9 Hz, 6H). ^13^C NMR (126 MHz, DMSO-*d6*) δ 166.42, 150.45, 149.91, 147.80, 147.57, 147.34, 146.81, 145.97, 127.74, 127.38, 126.33, 113.63, 112.33, 112.16, 111.78, 109.93, 103.57, 33.56, 24.01. HRMS calcd. for C_19_H_15_Cl_2_F_2_N_4_O_2_ ([M-H]^−^), 439.0540; found, 439.0550.

Compound **V-21** 4-Amino-3,5-dichloro-6-(5-(4-isopropylphenyl)-3-(trifluoromethyl)-1H-pyrazol-1-yl)picolinic acid. Yellow solid. ^1^H NMR (500 MHz, DMSO-*d6*) δ 13.98 (s, 1H), 7.43 (s, 2H), 7.34–7.21 (m, 5H), 2.87 (hept, *J* = 6.9 Hz, 1H), 1.17 (d, *J* = 6.9 Hz, 5H). ^13^C NMR (126 MHz, DMSO-*d6*) δ 165.68, 150.89, 150.40, 147.31, 146.54, 146.48, 143.24, 142.94, 142.64, 142.34, 127.88, 127.47, 125.64, 124.88, 122.74, 120.60, 118.46, 113.28, 112.90, 104.60, 33.59, 23.97. HRMS calcd. for C_19_H_14_Cl_2_F_3_N_4_O_2_ ([M-H]^−^), 457.0446; found, 457.0461.

Compound **V-22** 4-Amino-3,5-dichloro-6-(5-(3,4-dichlorophenyl)-3-methyl-1H-pyrazol-1-yl)picolinic acid. White solid. ^1^H NMR (500 MHz, DMSO-*d6*) δ 13.89 (s, 1H), 7.62 (d, *J* = 8.4 Hz, 1H), 7.55 (d, *J* = 2.0 Hz, 1H), 7.27 (s, 2H), 7.05 (dd, *J* = 8.4, 2.1 Hz, 1H), 6.68 (s, 1H), 2.27 (s, 3H). ^13^C NMR (126 MHz, DMSO-*d6*) δ 165.86, 150.71, 149.88, 147.37, 147.00, 142.49, 131.98, 131.58, 131.50, 130.40, 129.41, 127.30, 112.65, 112.39, 107.61, 13.75. HRMS calcd. for C_16_H_9_Cl_4_N_4_O_2_ ([M-H]^−^), 430.9450; found, 430.9460.

Compound **V-23** 4-Amino-3,5-dichloro-6-(5-(3,4-dichlorophenyl)-3-(difluoromethyl)-1H-pyrazol-1-yl)picolinic acid. White solid. ^1^H NMR (500 MHz, DMSO-*d6*) δ 14.03 (s, 1H), 7.69 (d, *J* = 2.1 Hz, 1H), 7.67 (d, *J* = 8.4 Hz, 1H), 7.42 (s, 2H), 7.23 (s, 1H), 7.14 (t, *J* = 54.2 Hz, 1H), 7.12 (dd, *J* = 8.4, 2.1 Hz, 1H). ^13^C NMR (126 MHz, DMSO-*d6*) δ 165.66, 150.96, 148.15, 147.92, 147.69, 147.41, 146.16, 143.52, 132.49, 132.21, 131.65, 130.02, 129.20, 127.62, 113.40, 113.12, 112.55, 111.54, 109.69, 105.15, 40.30. HRMS calcd. for C_16_H_7_Cl_4_F_2_N_4_O_2_ ([M-H]^−^), 466.9262; found, 466.9273.

Compound **V-24** 4-Amino-3,5-dichloro-6-(5-(3,4-dichlorophenyl)-3-(trifluoromethyl)-1H-pyrazol-1-yl)picolinic acid. White solid. ^1^H NMR (500 MHz, DMSO-*d6*) δ 14.11 (s, 1H), 7.74 (d, *J* = 2.1 Hz, 1H), 7.69 (d, *J* = 8.4 Hz, 1H), 7.48 (s, 1H), 7.45 (s, 2H), 7.14 (dd, *J* = 8.4, 2.1 Hz, 1H). ^13^C NMR (126 MHz, DMSO-*d6*) δ 165.69, 151.02, 147.80, 145.74, 144.00, 143.38, 143.08, 142.78, 142.48, 132.90, 132.30, 131.73, 130.24, 128.55, 127.70, 124.69, 122.55, 120.41, 118.27, 113.33, 112.39, 105.99. HRMS calcd. for C_16_H_6_Cl_4_F_3_N_4_O_2_ ([M-H]^−^), 484.9176; found, 484.9181.

Compound **V-25** 4-Amino-3,5-dichloro-6-(5-(4-ethylphenyl)-3-methyl-1H-pyrazol-1-yl)picolinic acid. White solid. ^1^H NMR (500 MHz, DMSO-*d6*) δ 13.22 (s, 1H),7.21 (s, 2H), 7.17 (d, *J* = 8.2 Hz, 2H), 7.12 (d, *J* = 8.2 Hz, 2H), 6.48 (s, 1H), 2.56 (q, *J* = 7.6 Hz, 2H), 1.14 (t, *J* = 7.6 Hz, 3H). ^13^C NMR (126 MHz, DMSO-*d6*) δ 165.94, 150.51, 149.52, 147.72, 147.12, 145.09, 144.55, 128.60, 127.39, 127.34, 113.15, 112.27, 106.21, 28.22, 15.58, 13.78. HRMS calcd. for C_18_H_15_Cl_2_N_4_O_2_ ([M-H]^−^), 389.0572; found, 389.0584.

Compound **V-26** 4-Amino-3,5-dichloro-6-(3-(difluoromethyl)-5-(4-ethylphenyl)-1H-pyrazol-1-yl)picolinic acid. White solid. ^1^H NMR (500 MHz, DMSO-*d6*) δ 14.05 (s, 1H), 7.37 (s, 2H), 7.23 (d, *J* = 8.4 Hz, 2H), 7.20 (d, *J* = 8.4 Hz, 2H), 7.11 (t, *J* = 54.2 Hz, 1H), 7.00 (s, 1H), 2.59 (q, *J* = 7.6 Hz, 2H), 1.15 (t, *J* = 7.6 Hz, 3H). ^13^C NMR (126 MHz, DMSO) δ 166.37, 150.46, 147.82, 147.59, 147.36, 146.76, 146.06, 145.36, 128.76, 127.77, 126.23, 113.63, 112.37, 112.18, 111.78, 109.93, 103.54, 40.28, 28.24, 15.51. HRMS calcd. for C_18_H_13_Cl_2_F_2_N_4_O_2_ ([M-H]^−^), 425.0384; found, 425.0394.

Compound **V-27** 4-Amino-3,5-dichloro-6-(5-(4-ethylphenyl)-3-(trifluoromethyl)-1H-pyrazol-1-yl)picolinic acid. White solid. ^1^H NMR (500 MHz, DMSO-*d6*) δ 13.95 (s, 1H), 7.41 (s, 2H), 7.26 (s, 1H), 7.24 (d, *J* = 8.3 Hz, 2H), 7.22 (d, *J* = 8.3 Hz, 2H), 2.59 (q, *J* = 7.6 Hz, 2H), 1.15 (t, *J* = 7.6 Hz, 3H). ^13^C NMR (126 MHz, DMSO-*d6*) δ 165.67, 150.86, 147.27, 146.61, 146.40, 145.88, 143.23, 142.94, 142.64, 142.34, 128.86, 127.89, 125.50, 124.86, 122.72, 120.58, 118.44, 113.25, 112.85, 104.55, 28.23, 15.48. HRMS calcd. for C_18_H_12_Cl_2_F_3_N_4_O_2_ ([M-H]^−^), 443.0289; found, 443.0306.

Compound **V-28** 4-Amino-3,5-dichloro-6-(3-methyl-5-(4-propylphenyl)-1H-pyrazol-1-yl)picolinic acid. White solid. ^1^H NMR (500 MHz, DMSO-*d6*) δ 14.14 (s, 1H), 7.21 (s, 2H), 7.15 (d, *J* = 8.3 Hz, 2H), 7.11 (d, *J* = 8.3 Hz, 2H), 6.48 (s, 1H), 2.51 (t, *J* = 7.6 Hz, 2H), 2.26 (s, 3H), 1.61–1.49 (m, 2H), 0.85 (t, *J* = 7.3 Hz, 3H). ^13^C NMR (126 MHz, DMSO-*d6*) δ 165.94, 150.51, 149.52, 147.71, 147.12, 145.10, 142.99, 129.15, 127.37, 127.31, 113.11, 112.24, 106.20, 37.29, 24.17, 14.11, 13.79. HRMS calcd. for C_19_H_17_Cl_2_N_4_O_2_ ([M-H]^−^), 403.0729; found, 403.0742.

Compound **V-29** 4-Amino-3,5-dichloro-6-(3-(difluoromethyl)-5-(4-propylphenyl)-1H-pyrazol-1-yl)picolinic acid. White solid. ^1^H NMR (500 MHz, DMSO-*d6*) δ 14.18 (s, 1H), 7.25 (s, 2H), 7.19 (t, *J* = 1.7 Hz, 4H), 7.10 (t, *J* = 54.2 Hz, 1H), 6.99 (s, 1H), 2.51 (t, *J* = 7.6 Hz, 2H), 1.60–1.50 (m, 2H), 0.85 (t, *J* = 7.3 Hz, 3H). ^13^C NMR (126 MHz, DMSO-*d6*) δ 165.75, 150.74, 147.93, 147.71, 147.47, 147.31, 146.83, 146.11, 143.88, 129.34, 127.69, 126.20, 113.59, 112.88, 111.74, 109.89, 103.65, 37.28, 24.12, 14.10. HRMS calcd. for C_19_H_15_Cl_2_F_2_N_4_O_2_ ([M-H]^−^), 439.0540; found, 439.0554.

Compound **V-30** 4-Amino-3,5-dichloro-6-(5-(4-propylphenyl)-3-(trifluoromethyl)-1H-pyrazol-1-yl)picolinic acid. Yellow solid. ^1^H NMR (500 MHz, DMSO-*d6*) δ 14.13 (s, 1H), 7.42 (s, 2H), 7.26 (s, 1H), 7.22 (s, 4H), 2.57–2.50 (m, 2H), 1.56 (m, 2H), 0.86 (t, *J* = 7.3 Hz, 3H). ^13^C NMR (126 MHz, DMSO-*d6*) δ 165.66, 150.87, 147.28, 146.62, 146.41, 144.31, 143.23, 142.93, 142.64, 142.33, 129.41, 127.82, 125.56, 124.87, 122.74, 120.60, 118.46, 113.22, 112.83, 104.55, 37.28, 24.09, 14.07. HRMS calcd. for C_19_H_14_Cl_2_F_3_N_4_O_2_ ([M-H]^−^), 457.0446; found, 457.0459.

Compound **V-31** 4-Amino-6-(5-(4-(tert-butyl)phenyl)-3-methyl-1H-pyrazol-1-yl)-3,5-dichloropicolinic acid. White solid. ^1^H NMR (500 MHz, DMSO-*d6*) δ 13.89 (s, 1H), 7.36 (d, *J* = 8.4 Hz, 2H), 7.24 (s, 2H), 7.15 (d, *J* = 8.4 Hz, 2H), 6.49 (s, 1H), 2.25 (s, 3H), 1.24 (s, 9H). 13C NMR (126 MHz, DMSO) δ 165.94, 151.39, 150.54, 149.50, 147.82, 147.13, 144.88, 127.04, 126.08, 113.27, 112.35, 106.27, 40.14, 34.86, 31.39, 13.78. HRMS calcd. for C_20_H_19_Cl_2_N_4_O_2_ ([M-H]^−^), 417.0885; found, 417.0903.

Compound **V-32** 4-Amino-6-(5-(4-(tert-butyl)phenyl)-3-(difluoromethyl)-1H-pyrazol-1-yl)-3,5-dichloropicolinic acid. White solid. ^1^H NMR (500 MHz, DMSO-*d6*) δ 13.92 (s, 1H), 7.77 (s, 2H), 7.42 (d, *J* = 8.5 Hz, 2H), 7.23 (d, *J* = 8.5 Hz, 2H), 7.11 (t, *J* = 54.2 Hz, 1H), 7.05 (s, 1H), 1.25 (s, 9H). ^13^C NMR (126 MHz, DMSO-*d6*) δ 166.91, 152.13, 150.16, 147.67, 147.44, 147.21, 146.77, 145.76, 127.43, 126.25, 126.04, 113.66, 111.81, 111.78, 111.45, 109.97, 34.93, 31.35. HRMS calcd. for C_20_H_17_Cl_2_F_2_N_4_O_2_ ([M-H]^−^), 453.0697; found, 453.0713.

Compound **V-33** 4-Amino-6-(5-(4-(tert-butyl)phenyl)-3-(trifluoromethyl)-1H-pyrazol-1-yl)-3,5-dichloropicolinic acid. White solid. ^1^H NMR (500 MHz, DMSO-*d6*) δ 13.99 (s, 1H), 7.43 (d, *J* = 8.5 Hz, 2H), 7.32 (s, 2H), 7.27 (d, *J* = 8.6 Hz, 3H), 1.25 (s, 9H). ^13^C NMR (126 MHz, DMSO-*d6*) δ 166.52, 152.57, 150.42, 146.40, 146.30, 143.03, 142.73, 142.43, 142.13, 127.57, 126.35, 125.51, 125.37, 124.93, 122.79, 120.65, 118.52, 112.35, 111.71, 104.46, 34.97, 31.32. HRMS calcd. for C_20_H_16_Cl_2_F_3_N_4_O_2_ ([M-H]^−^), 471.0602; found, 471.0620.

### 3.3. Homology Modeling of AFB5

The crystal structure of *A. Thaliana* AFB5 protein has not yet been verified; therefore, we conducted homology modeling based on TIR1 as the template protein from Protein Data Bank (PDB) coded 3C6O: chain B. The sequence of AFB5 obtained from TAIR (https://www.arabidopsis.org/, accessed on 10 October 2022) has a similarity of 50.98% with that of TIR1; therefore, it is feasible to construct the AFB5 protein structure upon TIR1 through homology modeling. The AFB5 protein was evaluated using SAVES6.0 (SAVESv6.0—Structure Validation Server (ucla.edu)) and SWISS MODEL (https://swissmodel.expasy.org/assess/, accessed on 15 October 2022) after optimizing the individual residues of the modeled protein using MOE. As shown in Figure 10, the Ramachandran plot showed 96.59% of all backbone dihedral angles in favored areas; 95.61% of the residues had an averaged 3D-1D score of ≥0.2, and the number of non-bond interactions formed between pairs of different atomic types on the side chain in the 3.5 nm range overall quality factor was 88.59 (≥0.2). The QMEAN value was −2.62, whereas the GMQE value was 0.80, which verified that the AFB5 protein modeled using MOE is of good quality.

### 3.4. Molecular Docking

Two-dimensional structures of picolinic acid derivatives were generated using ChemDraw Professional 16.0; their configurations were minimized and protonated, and charge was applied to the 3D structures using MOE2020. Thereafter, the protein was protonated, charge was added, and 4.5 Å of water molecules near the pocket was eliminated before molecular docking. The docking pocket and key residues were reported by Calderón Villalobos, which were put forward to the dock. The side chains near the pocket were set as free rotation, and refinement was set as an induced fit. London dG and GBVI/WSA dG were used as the rescoring functions. A total of 500 conformations of each compound were generated to predict their best possible binding pose and output 20 tops-core optimum configurations, which could be browsed on MOE, balancing score, and key binding residues to choose one conformation and output as the final conformation.

### 3.5. Biological Assay

#### 3.5.1. Root Growth Assays to Quantify Compounds Activity

The designed and synthesized compounds possess skeleton 2-picolinic acid and could exhibit the same inhibition on *Arabidopsis thaliana* root growth as auxin. Therefore, they were assayed for their influence on *Arabidopsis thaliana* root growth to explore preliminary bioactivity and evaluate their IC_50_ values. *Arabidopsis thaliana* seeds were surface-sterilized and spotted onto 1/2× Murashige and Skoog medium containing 0.7% agar, 3% sucrose, and compounds at the indicated concentrations in petri dishes. Subsequently, the dishes were transferred to the dark at 4 °C for 48 h. After that, the dishes were placed vertically into the incubator for 7 d at 22 °C for 16 h:8 h (day/night). The root lengths of 7-day-old seedlings were measured using IMAGEJ after the images were acquired.

The inhibition rates of root growth were determined as follows:$$P=\frac{L_{0}-L}{L_{0}}\times 100\%,$$
where P denotes the inhibition rate, and L and L_0_ are the average length of the *A. thaliana* root in the presence of compounds and in untreated controls, respectively.

Determination of the IC_50_ values was performed using an Internet tool: MLA—”Quest Graph™ IC_50_ Calculator.” AAT Bioquest, Inc., 4 November 2022, https://www.aatbio.com/tools/ic50-calculator.

#### 3.5.2. 3D-QSAR

Thirty-seven compounds, including thirty-three target compounds and four Yang compounds, were used for the 3D-QSAR study. Their structures and inhibitory activities are listed in Table 2. To develop the CoMFA model, all the compounds mentioned were used as a training set based on the structural and bioactive diversity using the DIVERSITY function of SYBYL-X 2.0 (Tripos, Inc., St. Louis, MO, USA). The compound structures in the test set sufficiently represent the diversity of the entire dataset. A training set was used to construct the model.

All compounds in the training set were drawn in ChemDraw and superimposed and aligned on the maximum common substructure of 6-(5-aryl-substituted-pyrazolyl)-2-picolinic acid derivatives using the alignment function in SYBYL-X 2.0. Based on the hypothesis that the common alignment core contributes equally to the bioactivity of the compounds, the conformational angles for the maximum common substructures of the most active compound **V-7** as the template for the alignment were copied and applied to the remaining compounds in the whole dataset. Figure 8 shows the most active compound **V-7** as the template, and the structural alignment of the training set for the 3D-QSAR study.

CoMFA was performed using the Lennard-Jones potential for the steric field and the Coulombic potential for the electrostatic field in SYBYL-X 2.0. The aligned compounds were placed in a 3D cubic lattice with 2.0 Å grid spacing in the x, y, and z directions, and these potentials were determined for each compound. A sp3 carbon probe with a van der Waals radius of 1.52 Å and a point charge of +1.0 was used at each lattice point of the grid box for the calculation of the steric and electrostatic fields. To avoid overpower due to the large steric and electrostatic energy values, the default energy cutoff value was set at 30 kcal/mol. The attenuation coefficient was set to a default value of 0.3. Regression analysis was performed using the cross-validated PLS method. To develop the final model by performing a PLS analysis, the first run was conducted through the cross-validation to obtain the ONC, and thereafter, the final run yielded the non-cross-validated r^2^ value.

#### 3.5.3. Greenhouse Herbicidal Activity Assay

Herbicidal activities were evaluated at the Collaborative Innovation Center for Green Pesticides, Zhejiang A&F University (Hangzhou, China), and the soil for herbicidal activity tests was collected from a local planted field. All compounds were dissolved in 100% DMSO and thereafter diluted with 0.1% Tween-80 solution to obtain the appropriate concentrations before use. The pre- and post-emergence herbicidal activities of the compounds were evaluated against six weeds, including three dicotyledonous weeds: SG, DS, and EC, and three gramineous weeds: CA, AT, and AR. In a greenhouse, the weed seeds were planted in a plastic pot with a diameter of 8.0 cm at the mouth of the pot, covered with 0.2–0.5 cm of soil after seeding, and the bottom was watered and cultured in a plant culture room at a temperature of 13 ± 8 °C. For the post-emergence herbicidal activity assays, the spray was applied at the 2–4 leaf stage of the weeds. For the pre-emergence herbicidal activity assay, the spray was applied 24 h after weed seeding. After the weed treatments, the weeds were transferred to a greenhouse for cultivation with standard management. Weed growth and toxic symptoms were observed regularly after treatment, and weed inhibitory activities were visually evaluated (two duplicates per experiment) 20 days after treatment to obtain the percent visual injury value. In the further herbicidal activity assay of compound V-8, fresh weight inhibition (three triplicates per experiment) in the aboveground was applied rather than visual observation. All sprays were performed using a Biotest Spray Tower (3WPSH-500E, Nanjing Institute of Agricultural Mechanization, Ministry of Agriculture and Rural Affairs).

#### 3.5.4. Crop Selectivity

In the greenhouse, three representative crops (corn, wheat, and sorghum) were used to evaluate the crop selectivity of the compounds using the procedure described above.

## 4. Conclusions

In this study, 33 4-amino-3,5-dicholor-6-(5-aryl-substituted-1-pyrazolyl)-2-picolinic acid compounds were designed and synthesized via a four-step synthetic route with good yields based on the splicing of the active fragments, wherein the phenyl-substituted pyrazolyl replaced the chlorine atom at position 6 of picloram. The docking analysis demonstrated that some compounds might be bioactive owing to their tighter affinity to AFB5 of *A. thaliana*. The primary bioassay for inhibiting *A. thaliana* root growth demonstrated that the IC_50_ value of compound **V-7** was 45-fold less than that of the commercial picloram herbicide. Based on this, a 3D-QSAR model was constructed and fitted well to the relationship between the bioactivity and structure, which could be used to design new lead compounds. The herbicidal activity tested in greenhouses demonstrated that compound **V-8** exhibited better post-emergence herbicidal activity against broadleaf weeds at a dosage of 300 gha^−1^ than picloram. A crop selectivity test indicated that compound **V-8** exhibited excellent crop safety against corn, wheat, and sorghum at a dosage of 300 gha^−1^ under post-emergence conditions. Nevertheless, the actual herbicidal activity of the compounds was different from the primary bioactive results, while the docking analysis was fit to *A. thaliana* root growth assays, which may be due to the biological differences between weeds and *A. thaliana*; therefore, the results were acceptable. These results demonstrated that the replacement of the chlorine atom at position 6 of picloram by phenyl-substituted pyrazolyl is favorable for improving the herbicidal activity of skeleton picloram, and compound **V-8** might be a potential lead structure for the discovery of novel synthetic auxin herbicides. Furthermore, these results provide new perspectives and insights for the future design of compounds with similar structures. Further studies on structural optimization and active mechanisms are currently in progress in our laboratory.

## Data Availability

Not applicable.

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
