# Peer review of "Design, Synthesis, Herbicidal Activity, and Structure–Activity Relationship Study of Novel 6-(5-Aryl-Substituted-1-Pyrazolyl)-2-Picolinic Acid as Potential Herbicides"

_molecules, 2023, doi:10.3390/molecules28031431_

Round 1
Reviewer 1 Report
The authors Feng et al. of the manuscript “Design, Synthesis, Herbicidal Activity, and Structure-activity Relationship Study of Novel 6-(5-Aryl-substituted-1-pyra-3 zolyl)-2-picolinic Acid as Potential Herbicides” underlined the potential of some compounds as novel synthetic auxin herbicides. This seems of a great importance since the experiment is well conducted and the manuscript is written following a logical order. A few minor changes and clarifications are required before publishing:
Introduction
I suggest shortening the introduction, some paragraphs are too long and many details are written. Please review this section and mention the information that are really needed there.
Results and Discussion
Line 127: Indicate Scheme 1. as numbered figure and then make the proper corrections for the other figures and the manuscript body.
Line 133-134: This statement “the target compound V exhibited a higher affinity for AFB5. In particular, the 133 binding energy of compound V-7 (-8.59 kJ mol-1) was the lowest.” is not clear. Please, rewrite it
Line 192: Please, add the reference number after “Golbraikh and Tropsha” and include it in the references list
Line 251: “V-1–V-18“ are two compounds so rewrite them as “V-1, V-18”
Line 252: Please, write the two compounds as “V-19, V-33”
Line 269: Please, delete the column “compound” from Table 5 and Table 6 since the compound is already indicated in table title.
Materials and Methods
Line 595: Check the numbering of the Figure!
Author Response
Response to Reviewer 1 Comments
Point 1:
Introduction
I suggest shortening the introduction, some paragraphs are too long and many details are written. Please review this section and mention the information that are really needed there.
Response 1:
We have shortened the introduction, and you can see more details in the attachment.
Point 2:
Results and Discussion
Line 127: Indicate Scheme 1. as numbered figure and then make the proper corrections for the other figures and the manuscript body.
Line 133-134: This statement “the target compound V exhibited a higher affinity for AFB5. In particular, the 133 binding energy of compound V-7 (-8.59 kJ mol-1) was the lowest.” is not clear. Please, rewrite it
Line 192: Please, add the reference number after “Golbraikh and Tropsha” and include it in the references list
Line 251: “V-1–V-18“ are two compounds so rewrite them as “V-1, V-18”
Line 252: Please, write the two compounds as “V-19, V-33”
Line 269: Please, delete the column “compound” from Table 5 and Table 6 since the compound is already indicated in table title.
Response 2:
Line 127: We have corrected the numbered figure.
Line 133-134: We have rewritten it.
Line 192: “Golbraikh and Tropsha conditions” is a general set of criteria for evaluation of predictive ability of QSAR models, so we think that the citation is unnecessary.
Line 251 & 252: “V-1–V-18” means from compound V-1 to compound V-18 instead of V-1 and V-18. It means 18 compounds instead of just two compounds. So as “V-19–V-33” in Line 252.
Line 269: We have deleted it.
Point 3:
Materials and Methods
Line 595: Check the numbering of the Figure!
Response 3:
Line 595: We have corrected the number of the figure.

Reviewer 2 Report
A very interesting comprehensive study, I made some line comments below and expressed some of my concerns.
Line 21 Change to “more intensively”
Line 34- Sentence is rambling, I would just state that arable land is limited based on crop production rates and population growth.
Line 40- Just say crops, planted crops is redundant
Line 48- Say “synthetic herbicides “as opposed to chemically synthesized
Line 53-54 Say structures as opposed to structural skeletons
Line 69 and Elsewhere- change launched as commercial herbicides to “Commercialized “
General Comments- What was the statistical design of the greenhouse experiment ? How many reps, what type of soil medium was used? Type of soil can significantly impact Pre herbicide activity, it can also influence POST activity if the herbicide is mobile through the xylem.
More statistical analysis is needed for the greenhouse results
Author Response
Response to Reviewer 2 Comments
Point 1:
Line 21 Change to “more intensively”
Response 1:
We have changed it.
Point 2:
Line 34- Sentence is rambling, I would just state that arable land is limited based on crop production rates and population growth.
Response 2:
We have revised the statement.
Point 3:
Line 40- Just say crops, planted crops is redundant
Response 3:
We have modified it.
Point 4:
Line 48- Say “synthetic herbicides “as opposed to chemically synthesized
Response 4:
We have modified it.
Point 5:
Line 53-54 Say structures as opposed to structural skeletons
Response 5:
We have modified it.
Point 6:
Line 69 and Elsewhere- change launched as commercial herbicides to “Commercialized “
Response 6:
We have modified it.
Point 7:
General Comments- What was the statistical design of the greenhouse experiment ? How many reps, what type of soil medium was used? Type of soil can significantly impact Pre herbicide activity, it can also influence POST activity if the herbicide is mobile through the xylem.
More statistical analysis is needed for the greenhouse results
Response 7:
Weed growth and toxic symptoms were observed regularly after treatment, and weed inhibitory activities was evaluated(two duplicates per experiment) visually 20 days after treatment to obtain percent visual injury value.
For compound V-8’s further herbicidal activity assay, evaluation (three triplicates per experiment) of fresh weight inhibition was applied in the aboveground against CA, AT, AR under pre-emergence (pre) and post-emergence (post) conditions.We used field soil from a planted field.
Tested soil was from a local planted field.
In Large-scale experiment, we used visual assessment to evaluate compound herbicidal activities by observing weed growth condition and toxic symptoms.
In further experiment for compound V-8, we used fresh weight inhibition to evaluate its herbicidal activity.
For more statistical analysis, we added a stacked bar chart to clearly indicate the biological activities of all compounds.
You can see more details in the attachment.
